# Significant Interference with Porcine Epidemic Diarrhea Virus Pandemic and Classical Strain Replication in Small-Intestine Epithelial Cells Using an shRNA Expression Vector

**DOI:** 10.3390/vaccines7040173

**Published:** 2019-11-02

**Authors:** Da Shi, Xiaobo Wang, Hongyan Shi, Jiyu Zhang, Yuru Han, Jianfei Chen, Xin Zhang, Jianbo Liu, Jialin Zhang, Zhaoyang Ji, Zhaoyang Jing, Li Feng

**Affiliations:** State Key Laboratory of Veterinary Biotechnology, Harbin Veterinary Research Institute, Chinese Academy of Agricultural Sciences, Xiangfang District, Haping Road 678, Harbin 150069, China; dashi198566@163.com (D.S.); wxb1901@163.com (X.W.); shy2005y@163.com (H.S.); 18841618894@163.com (J.Z.); hyr1968274346@163.com (Y.H.); chenjianfei@126.com (J.C.); zhangxin2410@163.com (X.Z.); liujianbo@caas.cn (J.L.); zhangjialin0106@gmail.com (J.Z.); Zy_ji2010@163.com (Z.J.); 15204604415@163.com (Z.J.)

**Keywords:** porcine epidemic diarrhea virus, RNA interference, processivity factor, intestine epithelial cells, *N* gene

## Abstract

Porcine epidemic diarrhea (PED) re-emerged in China in 2010 and is now widespread. Evidence indicates that highly virulent porcine epidemic diarrhea virus (PEDV) strains belonging to genotype G2 caused a large-scale outbreak of diarrhea. Currently, vaccines derived from PEDV classical strains do not effectively prevent infection by virulent PEDV strains, and no specific drug is available to treat the disease. RNA interference (RNAi) is a novel and effective way to cure a wide range of viruses. We constructed three short hairpin RNA (shRNA)-expressing plasmids (shR-N307, shR-N463, and shR-N1071) directed against nucleocapsid (N) and determined their antiviral activities in intestine epithelial cells infected with a classical CV777 strain and LNCT2. We verified that shR-N307, shR-N463, and shR-N1071 effectively inhibited the expression of the transfected *N* gene in vitro, comparable to the control shRNA. We further demonstrated the shRNAs markedly reduced PEDV CV777 and LNCT2 replication upon downregulation of N production. Therefore, this study provides a new strategy for the design of antiviral methods against coronaviruses by targeting their processivity factors.

## 1. Introduction

Porcine epidemic diarrhea (PED), caused by porcine epidemic diarrhea virus (PEDV), is a highly contagious intestinal infectious disease. PED is an important disease in swine-producing countries. PED causes the death of newborn piglets and weight loss in pigs of all ages from PEDV-induced severe symptoms, such as serious diarrhea, vomiting, and dehydration, which seriously damage the swine industry [1]. Following reports in 1978 [1], PED had an outbreak in swine-farming countries in Asia, North America, South America, and Europe [2,3,4,5]. Starting from the end of 2010, highly virulent PEDV variants that differed from the classic European strain CV777 were widespread in China, resulting in high mortality of newborn piglets and huge economic losses [6,7,8,9]. PEDV can be divided into genotypes G1 and G2 based on phylogenetic analysis of full-length *S* gene sequences [10]. PEDV strains detected in China since 2010 mostly belonged to genotype G2, which differed genetically from the CV777 vaccine strain belonging to subtype G1 [11]. However, many studies demonstrated that commercially available PEDV vaccines derived from classical strains of PEDV do not provide effective protection against highly virulent PEDV variant infections in China [3]. Thus, PEDV infection remains a major veterinary problem. Understanding it is essential to developing novel antiviral drugs that will specifically inhibit PEDV propagation.

RNA interference (RNAi) is a short, double-strand RNA-induced process that targets and degrades the messenger RNA (mRNA) of specific sequences [12,13]. Post-transcriptional gene silencing can be mediated by exogenous small interfering RNAs (siRNAs), endogenous microRNAs (miRNAs), and short hairpin RNAs (shRNAs). Effective gene knockdown is achieved by shRNAs by inducing the endogenous RNAi process [14,15]. Transcribed shRNAs are exported from the nucleus by Exportin-5 and processed by the RNase III Dicer into small double-stranded RNA (dsRNA) molecules of 19 to 23 bp called siRNAs. The complementary guide strand of processed siRNA is incorporated into the RNA-induced silencing complex to mediate the cleavage of target mRNAs [16,17]. RNAi evolves in the host defense system directed at infectious viruses and transposable elements. Effective silencing of transgene expression and endogenous genes in vivo was shown by a number of groups [6,7,18]. These findings raised the possibility that RNAi could be another therapeutic approach for inhibiting virus infection. ShRNAs were employed for therapy against human viral diseases, as well as cancer and neurogenerative diseases [8]. PEDV primarily infects villous epithelial cells throughout the small intestine and causes serious injury of intestine epithelial cells (IECs), including superficial villous enterocyte swelling and severe diffuse atrophic enteritis [19]. Infection of PEDV variant strains in the field complicates the development of prophylactic and therapeutic strategies to protect suckling pigs from diarrhea. As an RNA virus, PEDV could be an ideal target for studying its biology and therapeutics using RNAi.

The PEDV nucleocapsid (N) protein, which is abundantly expressed in infected cells, has multiple functions. It is one of the structural proteins that forms complexes with genomic RNA, enhancing viral transcription and assembly. Therefore, the N protein may be a potential drug target for antiviral therapy against PEDV infection. In this study, we compared and analyzed *N* gene sequences from 25 different PEDV G1 and G2 isolates from different countries. Three novel shRNAs targeting conserved and unexploited regions in the *N* gene were tested for inhibition of PEDV CV777 and LNCT2 replication. Cell viability, viral titer, and protein expression were examined as indicators of the efficacy of targeted gene silencing by the shRNAs. All three shRNAs effectively inhibited PEDV replication and *N* expression of the G1 and G2 subtypes.

## 2. Materials and Methods

### 2.1. Viral Propagation and Titer Assays

Swine intestinal epithelial cells (IECs) were donated by Dr. Yanming Zhang (Northwest A&F University, China), and derived from the mid-jejunum of neonatal, unsuckled, one-day-old piglets. Primary intestinal epithelial cells were isolated by the tissue explant adherent method and purified by trypsin digestion with citric acid, forming an undifferentiated porcine intestinal epithelial cell line that was immortalized. Cells were maintained in a Dulbecco’s minimum essential medium (DMEM; Gibco, Thermo Fisher Scientific, Waltham, MA, USA)/nutrient mixture F-12 (Ham) (1:1) containing 10% heat-inactivated fetal bovine serum (FBS; Gibco), 5 ng/mL epidermal growth factor (Life Technologies, Carlsbad, CA, USA), 5 mg/mL insulin-transferring selenium supplements (Life Technologies), and 1% penicillin–streptomycin (Life Technologies). Cell culture media were changed every two days, and cells were passaged every 3–4 days by trypsinization with 0.25% trypsin–ethylenediaminetetraacetic acid (EDTA) (Life Technologies). Vero E6 (African green monkey kidney cells, American Type Culture Collection (ATCC)) were grown and maintained in DMEM supplemented with 10% heat-inactivated FBS and penicillin–streptomycin, and incubated at 37 °C with 5% CO_2_.

The PEDV G1 CV777 vaccine strain (GenBank accession: AF353511.1) was preserved at Harbin Veterinary Research Institute (Harbin, China). PEDV G2 strain LNCT2 (GenBank accession: KT323980.1) was isolated in Vero E6 cells in our laboratory. Vero E6 cells were cultured and used to amplify PEDV as previously described [9]. After 70% of virus-infected cells showed cytopathic effects (CPEs), cultures were collected for three freeze–thaw cycles. Viral titration used 96-well microplates with Vero E6 cells. Viral cultures were 10-fold serially diluted with virus replication medium containing trypsin (10 μg/mL). Confluent Vero E6 cells from microplates were washed three times with phosphate-buffered saline (PBS) and inoculated at 0.1 mL per well into eight wells. Following adsorption for 1 h at 37 °C, the inocula were removed, and cells were washed three times with PBS. Subsequently, 0.1 mL of fresh virus replication medium was transferred into each well, and cells were incubated 4–5 days at 37 °C. The 50% tissue culture infective dose (TCID_50_) was expressed as the reciprocal of the highest dilution showing CPE by the Reed and Muench method. Assays were performed in triplicate in three independent experiments.

### 2.2. Plasmid Construction

Plasmids expressing GFP-tagged N and myc-Tagged N were described previously [20]. The design of shRNAs targeting the PEDV strain LNCT2 genome *N* gene (Figure 1A) used methods from the literature [21] and the web-based Block-iT^TM^ RNAi Designer program [22]. A Basic Local Alignment Search Tool (BLAST) search [23] was performed to exclude possible homologous sequences. Three individual targeting sites were selected and chemically synthesized (Sangon Biotech, Shanghai, China) (Table 1). Control shRNA was designed at the same time to have no homology with PEDV or the IEC cell genome. All sequences were arranged as *Bbs*I + sense + loop + antisense + termination signal + *BamH*I and cloned into the pGPU6-Hygro vector to make the shRNA-expressing plasmids shR-N307, shR-N463, shR-N1071, and shR-NC (Figure 1B,C). Expression of siRNAs was driven by the U6 promoter.

### 2.3. Plasmid DNA Preparation

Plasmid DNA was transformed into electrocompetent DH5α *Escherichia coli* and purified with EndoFree Plasmid Maxi Kits (QIAGEN, Hilden, Germany).

### 2.4. Cell Transfection and Antiviral Activity In Vitro

We validated the inhibitory effects of the shRNAs against N by target gene expression in an in vitro transfection system. One day before transfection, IEC cells were seeded into 12-well plates at 5 × 10^4^ cells per well without antibiotics and grown at 37 °C overnight with 5% CO_2_. The 60–80% confluent cells were transiently transfected with the indicated plasmid using Lipofectamine 3000 (Invitrogen, Carlsbad, CA, USA), according to the manufacturer’s instructions. At 48 h after transfection, *N* expression was analyzed by Western blot or inverted fluorescence microscope. To examine the inhibitory effects of shRNAs against N on target gene expression during PEDV replication, IEC cells were transfected with or without 1 μg, 2 μg, or 4 of μg shR-N307, shR-N463, and shR-N1071 or 4 μg of shR-NC for 24 h and infected with 100 TCID_50_/mL PEDV strain CV777 or LNCT2. At 48 h post-infection (hpi), antigen slides were prepared to detect PEDV N protein expression by Western blot. In parallel experiments, cells from individual wells were collected and frozen and thawed twice, followed by centrifugation at low speed (1000× *g*), and supernatants were serially diluted, inoculated with Vero E6 cells, and titrated by the Reed and Muench method.

### 2.5. Sodium Dodecyl Sulfate Polyacrylamide Gel Electrophoresis and Western Blots

IEC cells grown in 12-well plates were transfected and infected as described. Cells were harvested at the indicated time points after transfection or virus infection, washed once with cold PBS, and lysed in radioimmunoprecipitation assay (RIPA) buffer (Sigma-Aldrich, St. Louis, MO, USA) to determine protein concentrations. Equal amounts of protein were subjected to sodium dodecyl sulfate polyacrylamide gel electrophoresis (SDS-PAGE) followed by blotting onto nitrocellulose membranes. Membranes were washed twice in Tris-buffered saline (TBS) and incubated 2 h in Superblock blocking buffer (ThermoFisher, Waltham, MA, USA). Membranes were incubated with anti-N monoclonal antibody (mAb) 3G2 (prepared by our laboratory, diluted 1:1000), anti-myc monoclonal (Sigma, diluted 1:1000) or anti-glyceraldehyde 3-phosphate dehydrogenase (GAPDH) monoclonal antibody (Sigma, diluted 1:10,000). Proteins were revealed by IRDye 800CW goat anti-mouse immunoglobulin G (IgG) (H + L) (1:10,000) (LiCor BioSciences, Lincoln, NE, USA), and blots were visualized using an Odyssey infrared imaging system (LiCor BioSciences).

### 2.6. Cell Viability Assays

Cell viability assays were performed using cell counting kit-8 (CCK-8) (CK04; Dojindo, Shanghai, China), according to the manufacturer’s protocol. In brief, IEC cells were seeded in 96-well plates at 10,000 per well and incubated at 37 °C for 24 h. Cells were transfected or not with shRNAs, and plates were incubated for 48 h before 10 μL of CCK-8 was added to wells for incubation for 2 h. Optical density at 450 nm was measured. Viability of treated cells was expressed as a percentage relative to untreated cells.

### 2.7. Immunofluorescence Assays

IEC cells were seeded in 12-well plates, and confluent cell monolayers were transfected with shR-N307, shR-N463 and shR-N1071, or shR-NC (4 μg) with Lipofectamine 3000 (Invitrogen) before PEDV infection. Cells were infected with PEDV strain CV777 or LNCT2 at 100 TCID_50_/mL. PEDV infection was analyzed using immunofluorescence assays (IFAs) at 48 hpi. Cells were fixed with 4% paraformaldehyde at 4 °C for 30 min and washed with PBS. Fixed cells were permeabilized with 0.2% Triton X-100 for 15 min at room temperature and blocked with blocking buffer (PBS with 5% bovine serum albumin (BSA)) for 2 h. Preparations were labeled with the mouse anti-PEDV N mAb (1:100 dilution) at 37 °C for 2 h followed by labeling with Alexa Fluor 488 goat anti-mouse IgG antibody (1:200 dilution) (ThermoFisher) for 1 h at 37 °C. Cell nuclei were stained with 4′,6-diamidino-2-phenylindole (DAPI) (0.05 μg/mL) (D9542; Sigma) for 15 min and analyzed using an AMG EVOS F1 florescence microscope.

### 2.8. Statistical Analysis

All statistical data are expressed as means ± standard deviation (SD) of three independent experiments and analyzed using the Student’s *t*-test. A *p*-value <0.05 was considered statistically significant.

## 3. Results

### 3.1. IEC Cells and PEDV Infection

Phylogenetic analysis of CV777 and LNCT2 was based on complete genomic sequences (Figure 2A). CV777 clustered with PEDV G1 genotype (classic strains), whereas LNCT2 clustered with PEDV G2 genotype (epidemic strains), with >98% nucleotide identity to these strains. To determine PEDV propagation in IEC cells, CPEs and IFAs were used to monitor after infection (Figure 2B). At 48 hpi, CV777 and LNCT2 caused similar CPEs in IEC cells, characterized by rounding, aggregation, and rupturing. The morphology of negative control cells remained unchanged and did not exhibit any signs of CPEs. Inoculation of CV777 and LNCT2 with IEC cells resulted in positive immunofluorescent staining with mAb that recognized the virus N protein. Controls were negative for immunostaining.

### 3.2. Selection of Targeted Sites for N Gene-Specific shRNAs

The prevalent strains that caused the outbreak of PEDV in China in 2010 and in North America in recent years belonged to the G2 genotype. PEDV is a positive-stranded RNA virus with higher mutation rates than DNA viruses [24]. Specific shRNAs were designed to target the conserved gene. We used the *N* gene of LNCT2 as the target sequence. Using nucleotide substitutions in the *N* gene and previous reports [25], we chose three unused regions that were well conserved in *N* genes among 25 isolates of PEDV G1 and G2 genogroups. The sites were at +307 to +327 nt, +463 to +483 nt, and +1071 to +1091 nt relative to the 5′ ATG initiation codon (Table 1).

### 3.3. Cell Viability Is Not Affected by Plasmids with shRNAs

ShRNAs induce large amounts of cell death when transfected at large volume, resulting in interference with experimental results. To detect if the three shRNAs we used led to cytotoxicity at 4 μg, IEC cells were seeded in 96-well microplates and transfected with shR-N307, shR-N463, shR-N1071, or shR-NC (4 μg) using Lipofectamine 3000 (Invitrogen), or transfected with transfection reagent alone as mock. After transfecting for 48 h, CCK-8 solution (10 μL) was added to wells, and plates were incubated at 37 °C for 2 h. Absorbance was measured at 450 nm using a microtiter plate reader (Bio-Rad, Hercules, CA, USA). Viable cells that were mock transfected with transfection reagent alone were the reference of 100% cell viability. ShR-N307, shR-N463, and shR-N1071 exhibited no obvious cytotoxicity in transfected Vero E6 and IEC cells at 4 μg (Figure 3).

### 3.4. Efficient Inhibition of PEDV Myc-N or AcGFP-N Expression by shRNA Expression Cassettes

We designed three shRNAs specifically targeting regions shown in Figure 1, named shR-N307, shR-N463, and shR-N1071. Western blots determined the inhibitory effects of the shRNAs on *N* gene expression. PMyc-N was co-transfected into IEC cells with increasing doses of shR-N307, shR-N463, or shR-N1071 using Lipofectamine 3000. At 48 h after transfection, N protein expression was analyzed by Western blots using mouse anti-Myc mAb. Western blots demonstrated that shR-N307, shR-N463, and shR-N1071 inhibited Myc-tagged N protein expression in a dose-dependent manner (Figure 4A).

The *N* gene was fused with the *AcGFP* gene to make pAcGFP-N. The effect of shRNAs on *N* gene expression was monitored by *AcGFP* expression. Plasmid pAcGFP-N was transfected into IEC cells alone or with the indicated concentrations of shR-N307, shR-N463, or shR-N1071 expression cassettes. At 48 h post-transfection, the effects of shRNA on enhanced GFP (EGFP) expression were monitored by fluorescence microscopy. The shR-N307, shR-N463, and shR-N1071 expression cassettes targeting *N* gene inhibited the expression of *AcGFP* to some extent compared with cells transfected with plasmid pAcGFP-N alone. The shR-N307, shR-N463, and shR-N1071 expression cassettes targeting the *N* sequence were similar, with weak fluorescence observed from cells transfected with 4 μg of shRNAs (Figure 4B).

### 3.5. ShRNA-Mediated Inhibition of N Expression and PEDV Production in Infected IEC Cells

To examine the inhibitory effects of shRNAs against N on target gene expression during CV777 and LNCT2 replication, IEC cells were transfected with indicated plasmids for 24 h and then infected with PEDV CV777 orLNCT2 strain (100 TCID_50_/mL). At 48 hpi, PEDV-infected cells were lysed with RIPA buffer containing 1 mM phenylmethylsulfonyl fluoride (PMSF), and Western blots were used to analyze *N* expression using mAb 3G2. Treatment with shR-N307, shR-N463, and shR-N1071 significantly reduced *N* expression compared to shR-NC- or mock-transfected IEC cells. The inhibitory effects of the three shRNAs on *N* expression were elevated with increased shRNA concentration (Figure 5A and Figure 6A). These results show that *N*-targeted shRNA efficiently inhibited target gene expression during PEDV replication.

To determine if downregulation of *N* expression by shRNAs decreased PEDV CV777 and LNCT2 strain replication, IEC cells were transfected with indicated plasmids for 24 h and infected with CV777 or LNCT2 (100 TCID_50_/mL). Cell cultures were collected at 48 hpi, and viral titers were determined. Transfection with shR-N307, shR-N463, or shR-N1071 significantly reduced PEDV CV777 and LNCT2 replication compared to shR-NC- or mock-transfected IEC cells (*p* < 0.05) (Figure 5B and Figure 6B). Although differences in inhibition of PEDV CV777 and LNCT2 strain replication induced by 1 μg, 2 μg, or 4 μg of shR-N307, shR-N463, and shR-N1071 were seen, PEDV replication gradually reduced with increased shRNA concentrations.

## 4. Discussion

In this study, we demonstrated that shRNAs against PEDV N protein broadly inhibited PEDV G1 and G2 strains. This report found that shRNAs inhibit swine coronavirus replication in epithelia. Most coronaviruses such as severe acute respiratory syndrome coronavirus (SARS-CoV), Middle East respiratory syndrome coronavirus (MERS-CoV), and PEDV infect epithelial cells in the respiratory and/or enteric tracts. The control of coronavirus diseases is important for human public health security. Further developing RNAi as a potential therapeutic agent against coronavirus infection is worthwhile.

The PEDV N protein is predominantly produced in susceptible cells, which makes it a major target for early and accurate diagnosis [26]. The N protein forms complexes with coronavirus genomic RNA and enhances the viral transcription and assembly. Because of the importance of N protein in viral replication and the coronavirus *N* gene long used as a major target for shRNA design, we focused on the inhibition of PEDV infection using RNAi targeting the *N* gene of PEDV and systematically evaluated suppression efficiency. Alterations of viral genomes such as nucleotide substitution, insertion, and deletion have the potential to decrease the inhibitory effect of shRNA. We analyzed 25 *N* gene sequences derived from PEDV genotypes G1 and G2 and only found single-nucleotide substitutions. Considering the nucleotide substitutions in the *N* gene, we chose three conserved and unexploited regions for the design of shRNAs. PEDV is primarily transmitted through the fecal–oral route and infects intestinal villous epithelial cells in vivo [27]. Current in vitro cell cultures of PEDV include Vero E6 cells, MARC-145 cells (another monkey kidney cell line), and HEK293 cells [28]. Most are non-porcine intestinal epithelial cells and, thus, not ideal in vitro cellular models for studying the interaction between PEDV infection and the host response due to interspecific variation. The IEC cell line represented a better model of normal porcine intestinal epithelium than transformed cell lines, and provided a unique opportunity to explore host–pathogen interactions in an in vitro system [29]. Using this cell model, we showed that shRNA expression did not affect the viability of IEC cells. Co-transfection of recombinant plasmid pMyc-N and different concentrations of shRNAs highlighted the success of the gene knockdown at the protein level. GFP was an important reporter for *N* gene expression. For example, if the *N* gene was silenced by the shRNAs, the translation of GFP protein in frame with the *N* gene in the recombinant plasmid would also be inhibited. This hypothesis was consistent with our findings. Our shRNAs shR-N307, shR-N463, and shR-N1071 strongly reduced GFP expression, and the inhibition was dose-dependent. In addition, the model system was exploited to further study the pathological functions of PEDV genes. We assessed the capacity of shR-N307, shR-N463, and shR-N1071 for inhibiting the gene and protein expressions of PEDV CV777 and LNCT2 strains in vitro. We observed that shR-N307, shR-N463, and shR-N1071 were able to knock down target gene expression of PEDV CV777 and LNCT2 strains at 48 hpi. We also showed that shR-N307, shR-N463, and shR-N1071 against the *N* gene suppressed PEDV CV777 and LNCT2 strain replication.

Currently, siRNA import into cells requires vectors such as plasmids and recombinant lentiviruses that express shRNAs efficiently and stably. As recombinant lentiviruses have a higher adaptability and replication ability in host cells, and as their genes can be integrated into cellular genomes, the silencing effect of lentiviruses is more efficient than plasmids [30]. However, this kind of gene integration may induce host genome mutations and cause cellular injury. In addition, excessive expression of shRNAs competitively inhibits cell endogenous miRNAs and causes cytotoxicity [31]. Therefore, RNAi mediated by a plasmid is currently safer for high gene silencing efficiency. However, off-target effects from using plasmid-based siRNA are still a shortfall [32]. To solve this problem, designing improved siRNA sequences, as well as developing novel and progressive vectors, is necessary.

In conclusion, plasmids expressing three shRNAs targeting different sites of the *N* gene of PEDV were constructed and transfected into IEC cells. Infection of PEDV CV777 (G1) or LNCT2 (G2) occurred post transfection. Detection by Western blot and viral titer assays was used to measure levels of viral replication in cells. The results demonstrated that PEDV G1 and G2 strains were susceptible to RNAi pathways targeting the *N* gene. All shRNA tests led to the silencing of *N* gene expression and suppressed proliferation of PEDV G1 and G2 strains in IEC cells. In sum, our data showed the potential for the shRNA expression vectors to precisely and effectively interfere with the replication of PEDV G1 and G2 strains in vitro. To our knowledge, this is the first report of the inhibition of PEDV G1 and G2 strain infection with shRNAs in IEC cells. Therefore, this report enriches the antiviral spectrum of RNAi treatment. Although IEC cells are a non-transformed porcine intestinal epithelial cell line, these cells lack the complexity of the cell types found in the architecture of the intestinal epithelium and, thus, do not satisfactorily mimic the natural infection process. This method merits further investigation in animal studies to define its therapeutic potential. Determining if the technology could be used in vivo for anti-PEDV therapy is still under investigation.

## Figures and Tables

**Figure 1 vaccines-07-00173-f001:**
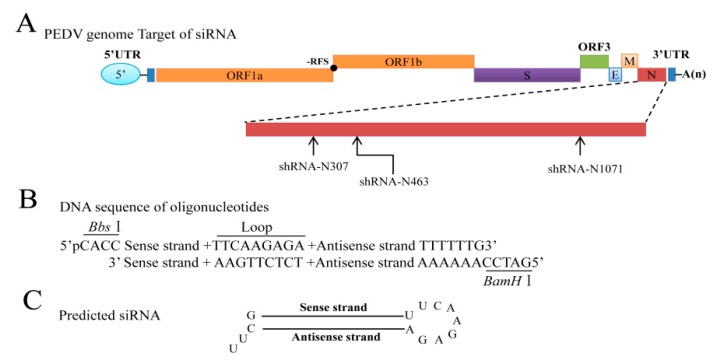
Schematic description of the target viral genome, and small interfering RNA (siRNA)-expressing cassette. (**A**) Genomic structure of porcine epidemic diarrhea virus (PEDV) and position of target short hairpin RNA (shRNA) at *N* gene. ShR-N307, shR-N463, and shR-N1071 indicate initial shRNA target sites; (**B**) sequences and design map for shRNA constructs; (**C**) structure of predicted shRNA.

**Figure 2 vaccines-07-00173-f002:**
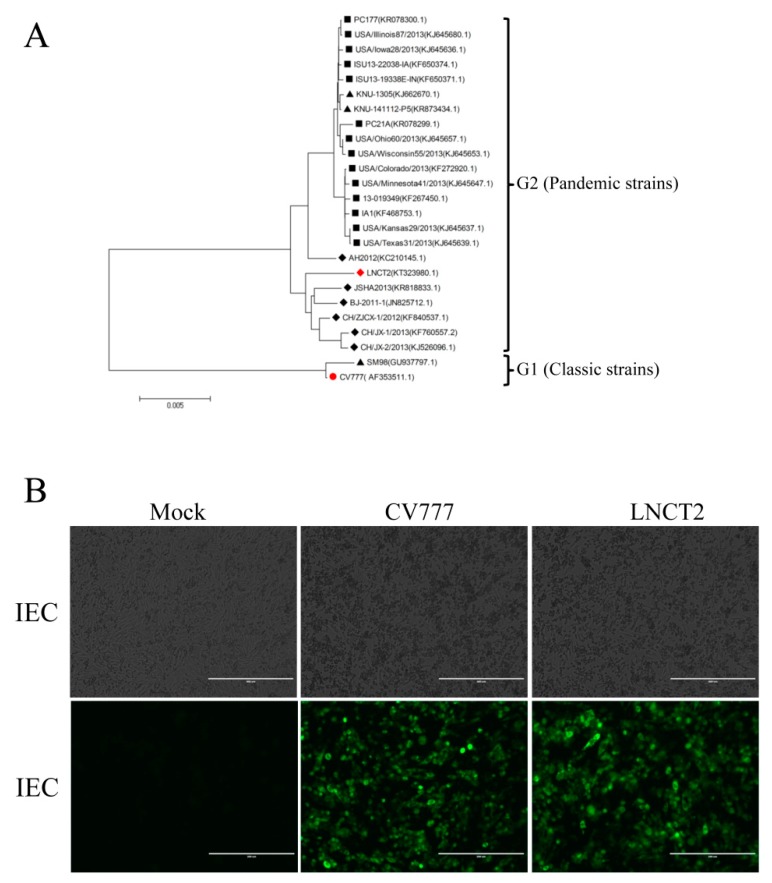
Phylogenetic analysis and virus-induced cytopathic effect (CPE) in intestine epithelial cells (IECs). (**A**) Phylogenetic trees of PEDV based on complete genomic DNA. Black squares, strains in United States of America (USA); black triangles, strains in South Korea; black rhombuses, strains in China; red circle, G1 CV777 strain from this study; red rhombus, G2 LNCT2 strain from this study. (**B**) Production and growth properties of PEDV CV777 and LNCT2 in IEC cells. CPE 48 h post-infection (hpi) (upper panels, bar: 400 µm) and immunofluorescence assay (IFA) 48 hpi (lower panels, bar: 200 µm) of CV777 and LNCT2 in IEC cells. IEC cells were infected with PEDV CV777 or LNCT2 at 100 50% tissue culture infective dose (TCID_50_)/mL. CPE and IFA were examined at 48 hpi, and cell images were captured.

**Figure 3 vaccines-07-00173-f003:**
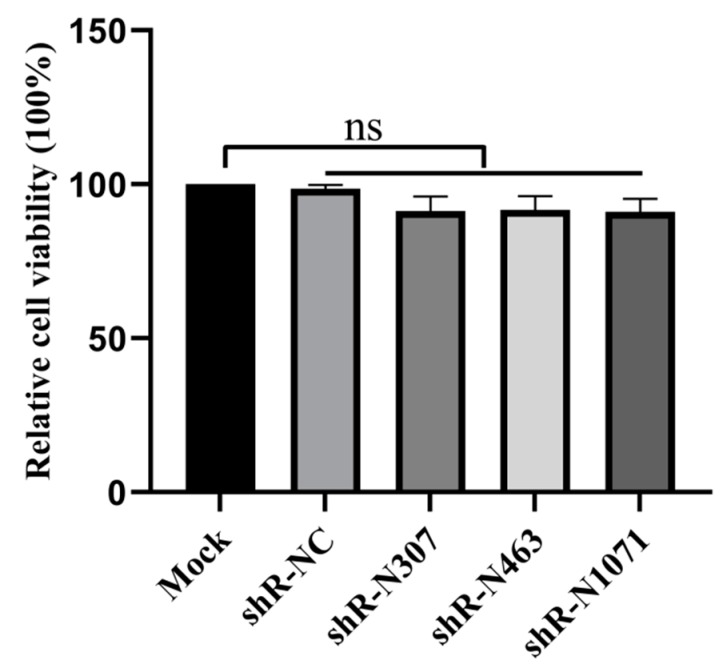
ShRNA treatment did not affect cell viability. Cell viability was detected by cell counting kit-8 (CCK-8) assays after transfection of IEC cells with shRNAs for 48 h. Absorption at 450 nm was recorded and expressed as a percentage of relative cell viability. Values are means ± SD (*n* = 3). Significant differences were assessed by Student’s *t*-test; ns, no significant difference compared to control; *p* > 0.05.

**Figure 4 vaccines-07-00173-f004:**
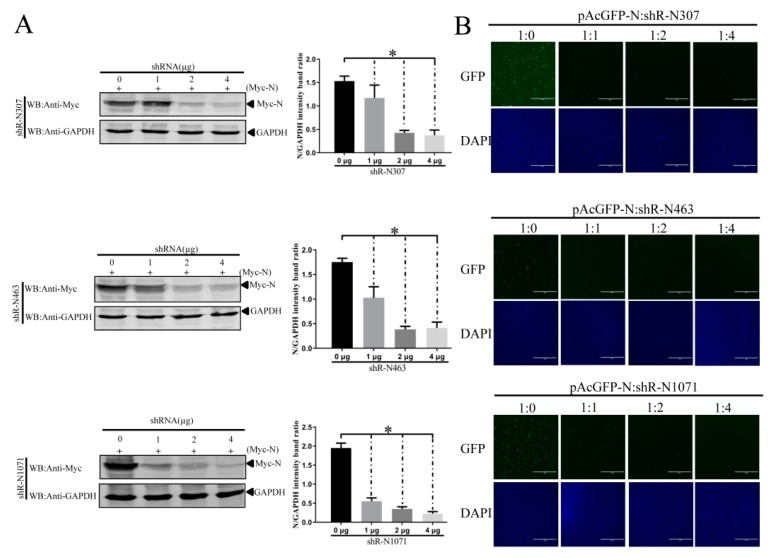
Inhibitory effects of shR-N307, shR-N463, and shR-N1071 on the PEDV N protein. (**A**) Western blots for effect of shR-N307, shR-N463, and shR-N1071 on *N* gene expression. PMyc-N was co-transfected with increasing doses of shR-N307, shR-N463, or shR-N1071 into IEC cells. After 48 h, transfected cells were lysed. Equal amounts of cell lysates were resolved by 12.5% SDS-PAGE. Reaction products were probed with anti-Myc or anti-glyceraldehyde 3-phosphate dehydrogenase (GAPDH). Densitometric data for N/GAPDH from three independent experiments are shown as means ± SD. * *p* < 0.05. The *p* value was calculated using Student’s *t*-test; (**B**) shR-N307, shR-N463, and shR-N1071 influence on pAcGFP-N expression in cultured IEC cells. PAcGFP-N was co-transfected with increasing doses of shR-N307, shR-N463, or shR-N1071 into IEC cells. PAcGFP-N expression plasmid was the unrelated control. Images show enhanced GFP (EGFP) expression at 48 h post-transfection.

**Figure 5 vaccines-07-00173-f005:**
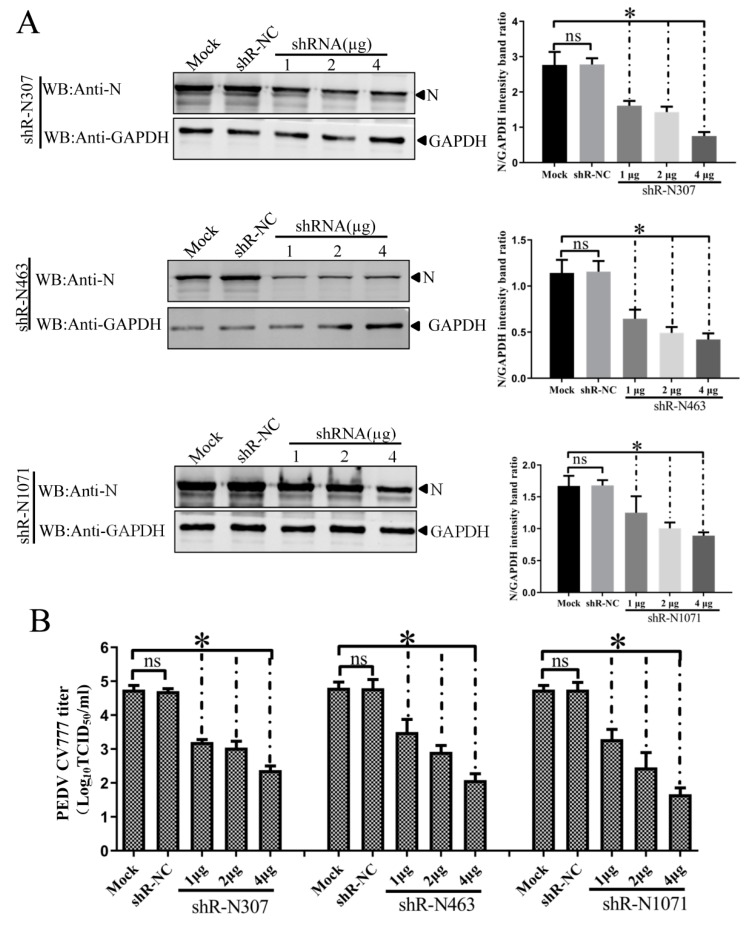
Dose-dependent inhibitory effects of shR-N307, shR-N463, and shR-N1071 against N on target gene expression and PEDV CV777 replication in infected IEC cells. IEC cells with shRNA (1, 2, 4 μg) for 24 h before infection with PEDV CV777 for 48 h. (**A**) PEDV N protein by Western blot with anti-N protein monoclonal antibody (mAb). Densitometric data for N/GAPDH from three independent experiments, shown as means ± SD. * *p* < 0.05. The *p* value was calculated using Student’s *t*-test; (**B**) PEDV CV777 titers in shRNA-transfected IEC cells. ShRNA transfection and viral infection were as in panel A. Viral titers in supernatants collected at 48 hpi were determined using the Reed–Muench method. Error bars represent standard errors of the mean from three independent experiments. * *p* < 0.05. The *p* value was calculated using Student’s *t*-test.

**Figure 6 vaccines-07-00173-f006:**
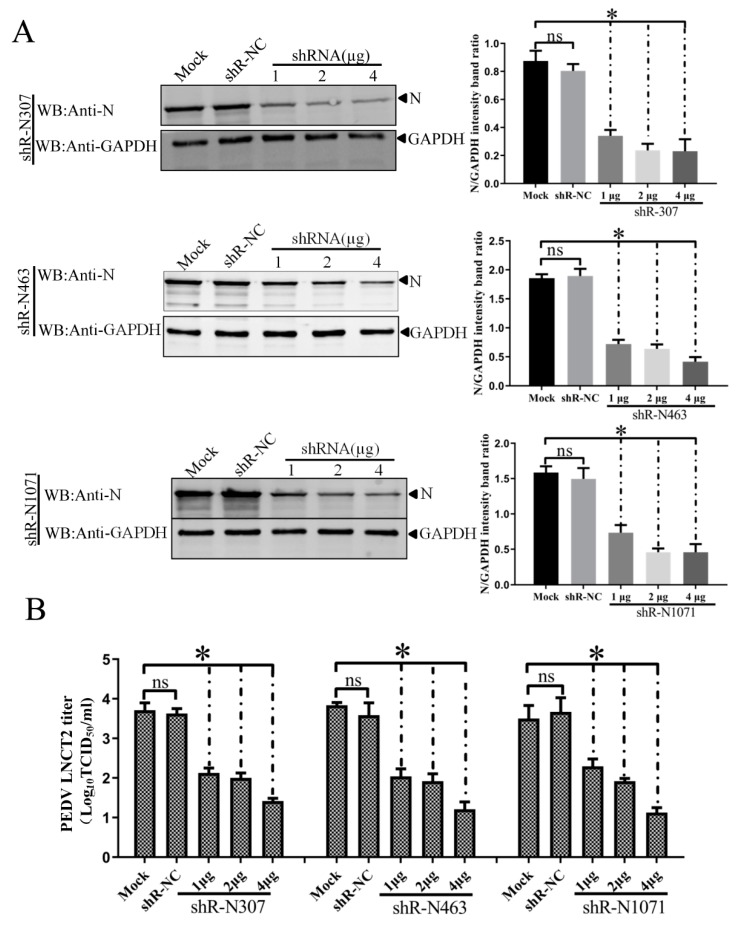
Dose-dependent inhibitory effects of shR-N307, shR-N463, and shR-N1071 directed against N on target gene expression and PEDV LNCT2 replication in infected IEC cells. IEC cells with shRNA (1, 2, 4 μg) for 24 h were infected with PEDV LNCT2 for 48 h. (**A**) PEDV N protein was detected by Western blot with anti-N protein mAb. Densitometric data for N/GAPDH from three independent experiments are shown as means ± SD. * *p* < 0.05. The *p* value was calculated using Student’s *t*-test; (**B**) PEDV LNCT2 titers in shRNA-transfected IEC cells. ShRNA transfection and viral infection were as in panel A. Viral titers in supernatants collected at 48 hpi were determined using the Reed–Muench method. Error bars represent standard errors of the mean from three independent experiments. * *p* < 0.05. The *p* value was calculated using Student’s *t*-test.

**Table 1 vaccines-07-00173-t001:** Gene sequence and position of RNA interference targets.

shRNA	Sequence	Position
shRNA-N307	GCAAAGACTGAACCCACTAAC	Position in *N* gene sequence: 307–327
shRNA-N463	GGCAACAACAGGTCCAGATCT	Position in *N* gene sequence: 463–483
shRNA-N1071	GCCAAAGTCTGATCCAAATGT	Position in *N* gene sequence: 1071–1091

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
