# Peer review of "Significant Interference with Porcine Epidemic Diarrhea Virus Pandemic and Classical Strain Replication in Small-Intestine Epithelial Cells Using an shRNA Expression Vector"

_vaccines, 2019, doi:10.3390/vaccines7040173_

Round 1

Reviewer 1 Report

In this manuscript a study on inhibition of N-protein production in swine intestinal epithelial cells by transfection with plasmids containing short hairpin RNA (shRNA) targeting the N protein gene from the LNCT2 genome in porcine epidemic diarrhea virus. The N protein was selected as target because of its multiple functions in viral transcription. After transfection with either one of three different plasmids targeting different locations in the protein N gene sequence, followed by infection with either one of two virus strains, LNCT2 representing an epidemic G2 genotype or CV777representing a classic G1 phenotype, reduction of protein N production was demonstrated in western blots and by reduced replication of the virus. These inhibitory activities were shown to be dependent on the concentration of the plasmids, and were not associated with cytotoxicity. It is concluded that the small interference RNA approach provides a useful antiviral method in infection with porcine epidemic diarrhea virus.

This is an interesting study demonstrating that the RNA interference methodology is effective in the inhibition of porcine epidemic diarrhea virus, being a coronavirus comprising a single stranded RNA genome. The experimental design is clear and the results are presented in a clear manner. There are ample controls included in the experiments. The combination of three different plasmids and two different viruses representing the two different genotypes underscores the value of the study.

There are a few suggestions which could improve the quality of the manuscript.

The first regards the use of intestinal epithelial cells (IEC). It is stated in the Materials and Methods that these are cells from a cell line, but further details are not given. Further clarifications are given in the Discussion “IEC cells were derived from jejunum epithelium from unsuckled 12-hour-old piglets, forming an undifferentiated porcine intestinal epithelial cell line that was immortalized. This line represented a better model of normal porcine intestinal epithelium than transformed cell lines [23].” It is suggested to bring this in the Materials and Methods with some more detail, and to focus the Discussion on the translational value of using cells from this cell line: noteworthy, it appears that cells from only one cell line were used in the study. At the end of the Conclusion it is written “In sum, our data showed the potential for the shRNAs expression vectors to precisely and effectively interfere with the replication of PEDV G1 and G2 strains in vitro.” This statement is correct, and it is advised to present some perspectives connected with this statement. In other words, is there a next step in further exploitation of the siRNA approach in the battle against infection by the virus and subsequent disease? This discussion should also consider the somewhat artificial approach in the in vitro studies, e.g., the use of cells from a cell line and the experimental approach in documenting the effect in in vitro. Also the observation that inhibition was not complete in the experiments should be discussed. It is advised to bring Supplemental Figure 1 in the main text. The use of English grammar is fine, but there are a few points for correction.

Reviewer 2 Report

This manuscript describes the results of in vitro experiments to reduce PEDV replication through short hairpin RNA transfection, with a view to developing future antiviral strategies.

Overall, the standard of English used must be improved in order to make the manuscript easier to understand.

The manuscript is well organised, and the figures are clearly presented and instructive.

Specific comments:

1- Lines 44-46 - The way this is written indicates that the vaccine itself is causing mortality - please re-word this sentence.

2- The source of many reagents, cells, etc. are not properly indicated.  For example, trypsin-EDTA (line 82), Vero E6 cells (line 83), viruses (line 85), monoclonal antibody (line 140), and others.

3- Lines 93-95 - It is not clear how the TCID50 was calculated - how were infected cells counted - through CPE, fluorescent staining, or otherwise?

4- Figure 2 - It would be useful in the figure legend to indicate the magnification of the image.
